# IL3 Has a Detrimental Effect on Hematopoietic Stem Cell Self-Renewal in Transplantation Settings

**DOI:** 10.3390/ijms232112736

**Published:** 2022-10-22

**Authors:** Parisa Tajer, Kirsten Canté-Barrett, Brigitta A. E. Naber, Sandra A. Vloemans, Marja C. J. A. van Eggermond, Marie-Louise van der Hoorn, Karin Pike-Overzet, Frank J. T. Staal

**Affiliations:** 1Department of Immunology, Leiden University Medical Center, 2333 ZA Leiden, The Netherlands; 2The Novo Nordisk Foundation Center for Stem Cell Medicine (reNEW), Leiden University Medical Center, 2333 ZA Leiden, The Netherlands; 3Department of Obstetrics, Leiden University Medical Center, 2333 ZA Leiden, The Netherlands

**Keywords:** IL3, hematopoietic stem cell, gene therapy, clinics, transplantation, ex vivo expansion

## Abstract

The ex vivo expansion and maintenance of long-term hematopoietic stem cells (LT-HSC) is crucial for stem cell-based gene therapy. A combination of stem cell factor (SCF), thrombopoietin (TPO), FLT3 ligand (FLT3) and interleukin 3 (IL3) cytokines has been commonly used in clinical settings for the expansion of CD34^+^ from different sources, prior to transplantation. To assess the effect of IL3 on repopulating capacity of cultured CD34^+^ cells, we employed the commonly used combination of STF, TPO and FILT3 with or without IL3. Expanded cells were transplanted into NSG mice, followed by secondary transplantation. Overall, this study shows that IL3 leads to lower human cell engraftment and repopulating capacity in NSG mice, suggesting a negative effect of IL3 on HSC self-renewal. We, therefore, recommend omitting IL3 from HSC-based gene therapy protocols.

## 1. Introduction

Long-term hematopoietic stem cells (LT-HSCs) give rise to all blood cells during the lifetime of an individual, through a process termed hematopoiesis. HSCs have the unique capacity of self-renewal and multipotency, while progenitors have a more defined path for specific lineage development [1,2,3]. HSCs from different sources have been used in the stem cell and gene therapy field. However, the main challenge of the limited number of HSCs that can be enriched from patients remains. Therefore, ex vivo expansion or maintenance of HSCs has become crucial to have long-term clinical benefits of transplanted cells. HSCs undergo symmetrical and asymmetrical cell divisions in vivo. Asymmetric division yields differentiating cells, whereas symmetric cell division leads to the expansion of HSCs in numerical terms. Therefore, approaches that will result in symmetric stem cell division and self-renewal without further differentiation are required for ex vivo expansion [4]. Different combinations of cytokines and additives have been assessed to expand HSCs in vitro; however, limited success has been reported in clinical settings, due to cell proliferation and cell cycle activation which subsequently leads to lower in vivo engraftment potentials of input cells [5,6,7,8]. Currently, a combination of stem cell factor (SCF), Flt3 ligand (FLT3), thrombopoietin (TPO) and interleukin 3 (IL3) are used as the key factors for the proliferation and maintenance of hematopoietic stem and progenitor cells (HSPCs) during ex vivo culturing systems in clinical settings [9,10]. Most clinical protocols for HSC-based gene therapy use these four cytokine cocktails, which we also use in our ongoing clinical trial for RAG1-SCID [11].

SCF and FLT3 were shown to induce essential signals for HSC development [12,13,14,15]; TPO has been known to stimulate the proliferation and expansion of HSCs [13]. On the other hand, controversial results have been reported for the effect of IL3 on the expansion of LT-HSCs. IL3 is known for its myelopoietic effect, the proliferation regulation of myeloid progenitors and the differentiation of granulocyte–monocyte progenitors into basophile in murine bone marrow [16,17,18,19]. Most of these studies could not confirm the clinical importance of IL3 in the long-term repopulation capacity of expanded LT-HSCs [20,21,22].

Therefore, we set out to evaluate the effect of IL3 on the long-term repopulating capacity of expanded HSCs in two cytokine combinations of SCF + TPO + FLT3 (STF) and SCF + TPO + FLT3 + IL3 (STF + IL3) for a short period of time (4 days), as is often used in gene therapy protocols. Our data show that IL3 strongly reduces the repopulation capacity of cultured hematopoietic stem cells.

## 2. Results

### 2.1. Ex Vivo Expansion of Hematopoietic Stem Cells in Cytokine-Supplemented Medium

To address the effect of IL3 on the ex vivo expansion of CD34^+^ cells, enriched CD34^+^ cells from cord blood were cultured in an X-vivo medium supplemented with a clinically used cytokine cocktail STF + IL3 (SCF, TPO, FLT3, and IL3) and STF (SCF, TPO and FLT3) for 4 days.

CD34^+^ are known to constitute a heterogenous population [23], therefore we performed extensive flow cytometry analysis to identify different subsets of hematopoietic stem cells and progenitors (HSPCs) at the start of culturing (referred to as day 0) and after 4 days of culture. The compositions of different subsets of HSPCs were identified by flow cytometry analysis based on well-known markers [24] (Appendix A).

Total nucleated cells (TNCs) during 4 days of culture in STF and STF + IL3 cytokine cocktail were expanded three and eightfold, respectively (Figure 1B). Although the increase in TNC was achieved during 4 days of culture, LT-HSCs were significantly lost in both cytokine cocktails (Figure 1C, Appendix A). On the other hand, other progenitors, specifically multi-lymphoid progenitors (MLPs), common myeloid progenitors (CMP) and granulocyte–macrophage progenitor (GMP) were expanded in both culture conditions compared to day 0 (Figure 1D, Appendix A). Overall, the addition of IL3 to the STF cytokine cocktail appears to benefit the numerical expansion of hematopoietic progenitors in vitro.

### 2.2. IL3 Reduces Human Engraftment in NSG Mice

To assess the repopulating capacity of expanded cells in vivo, cultured cells in both conditions were transplanted into NSG mice. The NSG mouse model is currently the accepted gold standard for assessing multilineage potential and self-renewal of human HSCs, because reliable in vitro assays to address stem cell functionality are absent. 

Peripheral blood (PB) was collected from week 4 to week 20 to monitor human engraftment over time. Mice were sacrificed at week 20 post-transplantation; peripheral blood, bone marrow (BM), thymus (Thy), and spleen (Sp) were collected and human engraftment (huCD45^+^) was assessed (Figure 2A). Mice that received expanded cells in an STF-supplemented medium show significantly higher human chimerism (% huCD45^+^) in PB compared to the mice that received cells expanded in STF + IL3 (Figure 2B). Flow plots of harvested BM cells show significantly higher chimerism and absolute count of huCD45^+^ cells in the STF group in comparison with the STF + IL3 group (Figure 2C,D). Consistently lower human chimerism in STF + IL3 group was also observed in the spleen and thymus (Figure 2E). BM cells of engrafted mice were further analyzed for the presence of mature hematopoietic cell populations. Cells within the huCD45^+^ gate were analyzed for myeloid, lymphoid and CD34^+^ populations (Appendix A).

Mice from the STF + IL3 group show lower percentages of CD34^+^ and CD3^+^ cells within CD45^+^ BM cells in contrast to the STF group, while no significant changes in other lineages were observed between the two groups (Figure 2F). Although a lower percentage of CD34^+^ was observed in the STF + IL3 group, no clear difference in HSC and MPP between groups was observed. Similar analyses were performed for other organs at week 20 post-transplantation. No significant differences among populations were observed except for CD3^+^ and CD4^+^CD8^+^ DP thymocytes, which show a decrease in the STF + IL3 group (Appendix A). Therefore, IL3 addition during the expansion phase of CD34^+^ HSPCs results in substantially lower overall engraftment in NSG mice, whereas the composition of the different hematopoietic lineages remains largely unaffected.

### 2.3. IL3 Reduces the Repopulation Capacity of Human Hematopoietic Stem Cells in Mice

To further evaluate the effect of IL3 on the long-term repopulation capacity of LT-HSCs, secondary transplantation was performed. In this experiment, BM cells from primary recipients were pooled and transplanted into secondary recipient mice (Figure 3A). Extensive flow cytometry analysis was performed at the time of transplantation to assess the absolute count and percentages of human CD45^+^, CD34^+^, HSC and MPP transplanted into each mouse per group (Appendix A). 

Human engraftment was assessed in PB from week 4 up to 20 weeks post-transplantation. Additionally, after the secondary transplantation, IL3 addition led to a significant decrease in human chimerism in PB (Figure 3B) and similarly in BM, spleen and thymus (Figure 3C,D); this decrease is more pronounced than the primary transplantation. Cells within the huCD45^+^ gate were further analyzed for the presence of different lineages development in BM. IL3-treated stem cells differentiated much worse into CD19^+^, CD34^+^, and CD56^+^ cells, presumably due to lower self-renewal of HSC populations in BM (Figure 3E,F). 

Similar analyses were performed in the PB, spleen and thymus of secondary recipient mice 20 weeks post-transplantation. The group STF + IL3 showed significantly lower lymphoid development in PB, thymus and spleen, whereas the CD13^+^CD33^+^ myeloid population is unaffected (Appendix A). Taken together, IL3 affects not only human engraftment in NSG mice but reveals a specific defect in the reconstitution of the lymphoid compartment upon secondary transplantation.

## 3. Discussion

Different cytokine combinations and small molecules such as UM171, and SR1 have been studied for the expansion and maintenance of hematopoietic stem and progenitor cells in vitro [13,22,25,26,27]. Successful expansion of CD34^+^ cord blood using UM171 for allogenic stem cell transplantation for cancer patients has been achieved in clinical settings (www.clinicaltrials.gov, accessed on 29 January 2016, NCT02668315). 

However, some of the studies using only traditional cytokine combinations have failed to consider the clinical implications of cytokine combinations on expansion and engraftment of CD34^+^ progenitor cells, as mainly in vitro assays such as colony-forming assays or long-term culture initiating cell assays have been used to assess the functionality of LT-HSCs.

Based on early findings, combinations of stem cell factor (SCF), Flt3 ligand (FLT3), thrombopoietin (TPO) and interleukin 3 (IL3) have been established as crucial factors for supporting HSC and progenitors proliferation and maintenance during in vitro culture systems, particularly in clinical settings [9,10]. However, controversial results have been reported on the effect of IL3 on the ex vivo expansion of long-term hematopoietic stem cells. 

IL3 supports myelopoiesis and the proliferation of lineage-committed progenitors in culture [28,29]. Some studies reported negative effects of IL3 on the engraftment of expanded murine cells in the bone marrow of recipient mice [21,30]. Piacibello et al. showed lower engraftment of cultured human CD34^+^ cells with IL3; however, repopulating capacity of engrafted cells in secondary transplantation in NSG was not determined [20]. In this study, we assess the effect of IL3 on the maintenance of in vivo repopulating capacity of ex vivo expanded human CD34^+^ cells from cord blood. 

Collectively, our results from the primary and secondary transplantations in NSG mice show that ex vivo expanded CD34^+^ cells in the presence of IL3 fail to preserve their repopulation and self-renewal capacity, suggesting the negative effect of IL3 on the maintenance of LT-HSCs in culture. Moreover, the data from the secondary transplantations reveal the detrimental effect of IL3 on the long-term development of the lymphoid lineage (B, T, NK cells) in BM, PB, spleen and thymus.

The inclusion of IL3 in a clinical protocol for expansion of CD34^+^ for gene therapy purposes results in an increase in total CD34^+^ cells [11]; however, the quality of LT-HSCs is compromised in favor of the expansion of lineage-committed progenitors such as MLP, CMP and GMP. Following earlier studies, our findings emphasize the importance of the maintenance and expansion of LT-HSCs for gene and cell therapy purposes. Thus, optimal culture condition for ex vivo expansion of HSPCs that retain their self-renewal capacity is essential for HSC-based gene therapy, especially for diseases such as SCID, XLA and others that primarily target the lymphoid compartment. Based on these considerations we propose omitting IL3 for HSC-based gene therapy aimed at restoring the lymphoid lineages and thereby regenerating adaptive immunity.

## 4. Materials and Methods

### 4.1. Human Cells and CD34^+^ Enrichment

Human cord blood was obtained after informed consent from Leiden University Medical Center. Mononuclear Cells (MNCs) were obtained from cord blood by density centrifugation using Ficoll-Amidotrizoaat. CD34^+^ cells were positively selected using the human CD34 UltraPure MicroBead Kit (Miltenyi Biotec, Bergisch Gladbach, Germany) according to the manufacturer’s protocol. 

### 4.2. CD34^+^ Cell Culture

100,000 enriched CD34^+^ cells/mL were cultured in X-vivo15 medium (Lonza) supplemented with recombinant huSCF (300 ng/mL), huTPO (100 ng/mL), huFLT3 (100 ng/mL) and huIL3 (20 ng/mL) (from Miltenyi Biotec). After 4 days of culture, expanded cells were harvested and counted using a nucleocounter 3000 (Chemometec, Allerod, Denmark) for subsequent immunophenotyping and transplantation in mice.

### 4.3. Mice

NOD.Cg-Prkdcscid Il2rgtm1Wjl/SzJ (NSG) mice were purchased from Charles River (France). All animal experiments were approved by the Dutch Central Commission for Animal experimentation (Centrale Commissie Dierproeven, CCD).

### 4.4. Primary and Secondary Transplantations into NSG Mice

For the primary transplantations, ex vivo expanded CD34^+^ cells (50,000 total nucleated cells per mouse) in Iscove’s Modified Dulbecco’s Medium (IMDM) without phenol red (Gibco) were transplanted by tail vein injection into pre-conditioned recipient NSG mice (*n* = 5). For the secondary transplantations, bone marrow (BM) cells from primary recipient mice from each group were pooled and 1/7th of the pooled cells were injected into pre-conditioned secondary recipient NSG mice (*n* = 5). 6–8-week-old recipient mice were conditioned with two consecutive doses of 25 mg/kg Busulfan (Sigma-Aldrich) (48 h and 24 h prior to transplantation). Mice used for transplantation were kept under specific pathogen-free conditions. The first four weeks after transplantation mice were fed with additional DietGel recovery food (Clear H_2_O) and antibiotic water containing 0.07 mg/mL Polymixin B (Bupha Uitgeest), 0.0875 mg/mL Ciprofloxacin (Bayer b.v.). Peripheral blood (PB) from the mice was drawn by tail vein puncture every 4 weeks until the end of the experiment. At the end of the experiment, PB, thymus, spleen and BM were harvested. Mice were euthanized via CO_2_-asphyxiation.

### 4.5. Flowcytometry

Ex vivo cultured cells were stained with the live/dead marker Zombie (Biolegend) according to the manufacturer’s instructions. Subsequently, cells were stained with antibodies listed in Appendix A and incubated for 30 min at 4 °C in the dark in FACS Buffer (PBS pH 7.4, 0.1% azide, 0.2% BSA). Single-cell suspensions from murine BM, thymus and spleen were prepared by squeezing the organs through a 70 μM cell strainer (BD Falcon). Erythrocytes from PB and spleen were lysed in NH_4_Cl (8,4 g/L)/KHCO_3_ (1 g/L). Single-cell suspensions were counted and stained as described above. All cells were measured on an Aurora spectral flow cytometer (Cytek). The data were analyzed using FlowJo software (Tree Star).

### 4.6. Statistics

Statistics were calculated and graphs were generated using GraphPad Prism9 (GraphPad Software). Statistical significance was determined by standard one/two-tailed Mann–Whitney U tests, unpaired *t*-tests and ANOVA tests (* *p* < 0.05, ** *p* < 0.01, *** *p* < 0.001 and **** *p* < 0.0001).

## Figures and Tables

**Figure 1 ijms-23-12736-f001:**
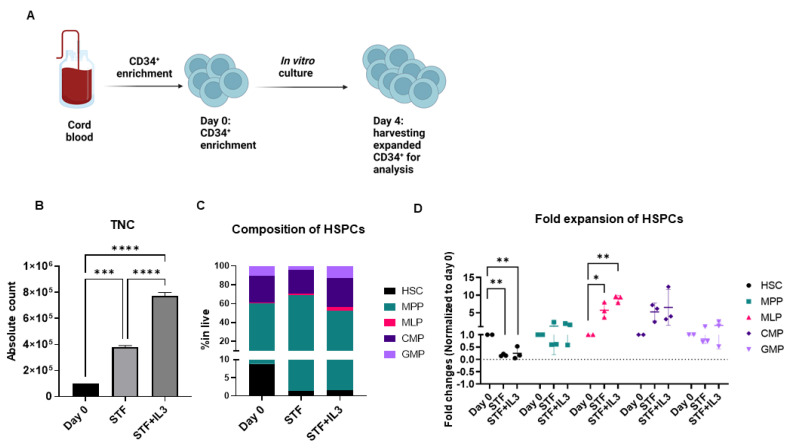
Human CD34^+^ cells were expanded in cytokine-supplemented medium: (**A**) Schematic experimental design. (**B**) Total Nucleated Cells (TNC) increased between three- and eightfold in presence of cytokine after 4 days of culture (*n* = 3, *** *p* < 0.001, **** *p* < 0.0001, one-way Anova). (**C**) Stacked bar graph shows composition of Hematopoietic Stem cells and Progenitors (HSPCs) on day 0 (after enrichment) and after 4 days of culture with the indicated cytokine cocktails. (**D**) Dot plot shows fold changes of expanded HSPCs on day 4 compared to day 0. (*n* = 3, * *p* < 0.05, ** *p* < 0.01, one-way Anova).

**Figure 2 ijms-23-12736-f002:**
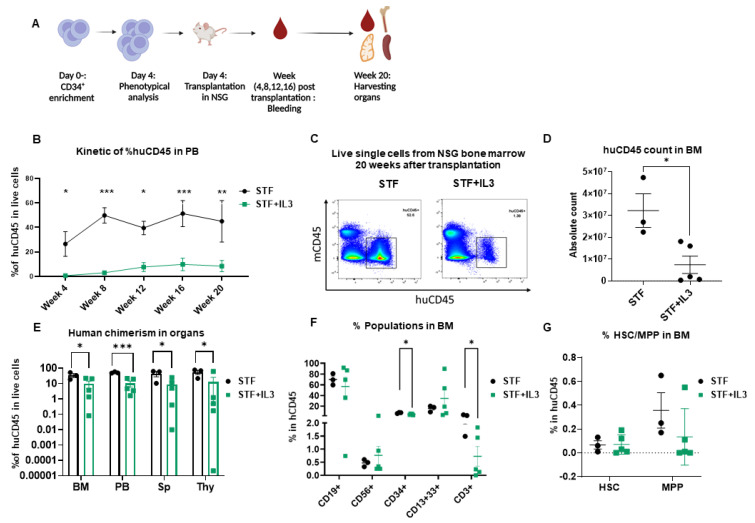
IL3 has an adverse effect on human engraftment in NSG mice: (**A**) Schematic experimental design. (**B**) Human CD45 engraftment in PB over time (*n* = 3–5; * *p* < 0.05, ** *p* < 0.01 and *** *p* < 0.001, 2-way Anova test with multiple comparison). (**C**) Representative flow cytometry plots of bone marrow cells harvested from mice that received expanded cells. (**D**) Absolute count of human CD45^+^ cells in bone marrow after 20 weeks of transplantation. (*n* = 3–5; * *p* < 0.05 unpaired *t*-test). (**E**) Percentage of human CD45 cell engraftment in different organs (BM = Bone marrow, PB = Peripheral Blood, Sp = Spleen, Thy = Thymus) in primary recipient NSG mice 20 weeks after transplantation (*n* = 3–5; * *p* < 0.05, and *** *p* < 0.001, unpaired *t*-test). (**F**) Percentage of CD19^+^, CD56^+^, CD34^+^, CD13^+^33^+^ and CD3^+^ populations within the huCD45^+^ BM population (*n* = 3–5; * *p* < 0.05, unpaired *t*-test). (**G**) Percentage of hematopoietic stem cells (HSC) and Multipotent progenitor (MPP) in BM of primary recipient NSG mice 20 weeks after transplantation (*n* = 3–5; unpaired *t*-test).

**Figure 3 ijms-23-12736-f003:**
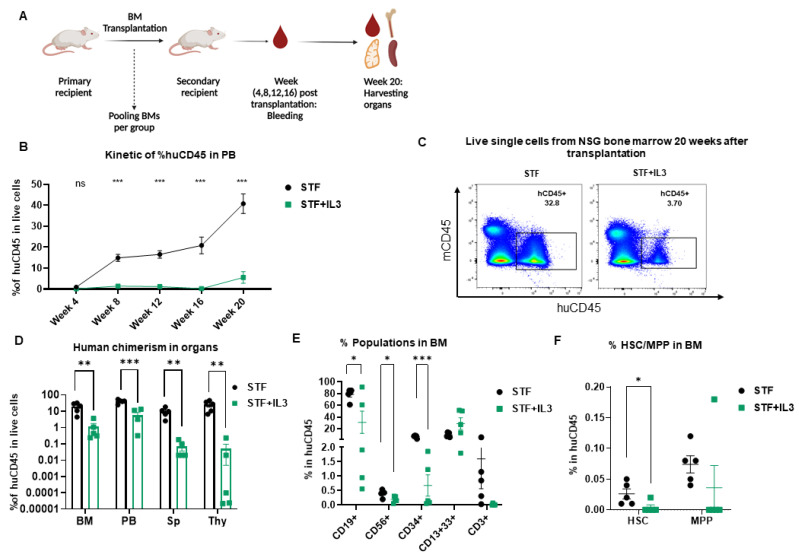
Negative effect of IL3 on repopulating capacity of hematopoietic stem cells revealed by secondary transplantation: (**A**) Schematic picture of secondary transplantation. (**B**) Human engraftment in PB (*n* = 5; not significant (ns) > 0.99, *** *p* < 0.001, 2-way Anova test with multiple comparison). (**C**) Flow plots of harvested BM cells show human engraftment. (**D**) Percentage of human CD45 cells in different organs (BM = Bone marrow, PB = Peripheral Blood, Sp = Spleen, Thy = Thymus) from secondary recipient NSG mice 20 weeks after transplantation (*n* = 5; ** *p* < 0.01 and *** *p* < 0.001, unpaired *t*-test). (**E**) Percentage of CD19^+^, CD56^+^, CD3^+^, CD34^+^ and CD13^+^33^+^ populations within the huCD45^+^ BM population from secondary recipient NSG mice 20 weeks post-transplantation (*n* = 5; * *p* < 0.05, and *** *p* < 0.001, unpaired *t*-test). (**F**) Percentage of hematopoietic stem cells (HSC) and Multipotent progenitor (MPP) in BM of primary recipient NSG mice 20 weeks after transplantation (*n* = 5; * *p* < 0.05, unpaired *t*-test).

## Data Availability

The data presented in this study are available on request from the corresponding author.

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
