# Peer review of "IL3 Has a Detrimental Effect on Hematopoietic Stem Cell Self-Renewal in Transplantation Settings"

_ijms, 2022, doi:10.3390/ijms232112736_

Round 1
Reviewer 1 Report
The study by Tajer et al demonstrates that the cytokine IL3, which is commonly included in cultures of human CD34+ HSPCs, promotes the loss of cells with the properties of long-term HSCs.
- The authors mention in the Introduction that the effects of IL3 on LT-HSCs are controversial. No discussion of IL3 concentration is included. Is IL3 concentration in the cultures standard in the current study and all previous controversial studies? Do the authors expect that IL3 will have the same detrimental effect on LT-HSCs across a range of concentrations?
- Further discussion of other experimental CD34+ expansion protocols should be included, such as the use of UM171 and other experimental protocols that perform much better than the traditional TPO/SCF/FLT3 cultures. Do the authors recommend to omit IL3 from these experimental CD34+ expansion protocols also?
- In several of the graphs analyzing the composition of the huCD45+ cells in primary and secondary NSG recipients (%, Fig. 2F, 3E-F) most of the cell lineages are depleted in the mice reconstituted with CD34+ cells from the STF+IL3 cultures. If I understand correctly, these graphs represent % of total. Which cell lineage is than expanded in these mice relative to all others?
- Error bars on some of the graphs are missing e.g. 1B. To judge the reproducibility of the data it would be helpful for the authors to indicate if various experiment were conducted more than once, or if the results were reproduced across several independent experiments.
- Many of the Figures are somewhat blurry. Higher resolution is recommended to make the small fonts on the axis of the graphs easier to read. The flow cytometry plots in Supplemental S1A are very blurry.
- CD13-CD33+ cell population is analyzed as myeloid cells. Do the authors mean CD13+CD33+, rather than CD13-CD33+? Can you explain which subsets of myeloid cells is this population expected to include?
Author Response
- The authors mention in the Introduction that the effects of IL3 on LT-HSCs are controversial. No discussion of IL3 concentration is included. Is IL3 concentration in the cultures standard in the current study and all previous controversial studies? Do the authors expect that IL3 will have the same detrimental effect on LT-HSCs across a range of concentrations?
We thank the reviewer for pointing out this important issue. In this study we used the cytokine concentrations currently used in clinical settings, especially in clinical gene therapy programs for various types of SCID. Although, in terms of research settings, various concentrations of cytokines are being used in the field, which make it quite difficult to compare, as in our experience different concentration of each cytokines can have significant effect on expansion of CD34+, HSC, and other progenitors.
Consistence with our study, a study by Piacibello et al (2000), comparable concentrations of IL3 (20ng/ul) were used. Similarly, they have shown the negative effect of IL3 on engraftment of expanded CD34+ in NSG mice, although different concentration of SCF, TPO and FLT3 have been used in their study (Reference #20). Given the strong myeloid differentiation capacities of IL3, we anticipate that a wide range of IL3 levels will have a similar effect, but we cannot exclude lack of negative effects at much lower concentrations.
- Further discussion of other experimental CD34+ expansion protocols should be included, such as the use of UM171 and other experimental protocols that perform much better than the traditional TPO/SCF/FLT3 cultures. Do the authors recommend to omit IL3 from these experimental CD34+ expansion protocols also?
Many thanks for this useful and important addition. We have incorporated the suggested compounds into the discussion section in line 147-150. IL3 was not included in the cytokine combination for CD34+ expansion using UM171, done by Fares et al (reference #27). However, in another study by Ngom, IL3 was used in combination of UM171 for short term culture for LV transduction purposes. Based on the results of our study and Fares et al., we suggest that IL3 should be removed from these expansion protocols.
- In several of the graphs analyzing the composition of the huCD45+ cells in primary and secondary NSG recipients (%, Fig. 2F, 3E-F) most of the cell lineages are depleted in the mice reconstituted with CD34+ cells from the STF+IL3 cultures. If I understand correctly, these graphs represent % of total. Which cell lineage is than expanded in these mice relative to all others?
In figure 2F and 3E-F, we showed the percentage of several key populations (e.g., %CD19+, CD56+, CD34+,….) within huCD45+ cells. Given the very low engraftment in the STF+IL3 group, we observed a significant decrease in percentages of CD34+ and CD3+, however the majority of cells are CD13+CD33+ in BM. As an example, below you can find the same data from 2F in an stacked graph for better visualization.
- Error bars on some of the graphs are missing e.g. 1B. To judge the reproducibility of the data it would be helpful for the authors to indicate if various experiment were conducted more than once, or if the results were reproduced across several independent experiments.
We have adjusted these graphs. More data were included, and statistical tests were performed. See updated Figure1 and S1.
- Many of the Figures are somewhat blurry. Higher resolution is recommended to make the small fonts on the axis of the graphs easier to read. The flow cytometry plots in Supplemental S1A are very blurry.
All the pictures have now been exported as PNG. Adjustments have been made to increase their quality.
- CD13-CD33+ cell population is analyzed as myeloid cells. Do the authors mean CD13+CD33+, rather than CD13-CD33+? Can you explain which subsets of myeloid cells is this population expected to include?
Both CD13 and CD33 are myeloid associated antigens. We used the combination of CD13 and CD33 in a single fluorochrome (Table 1 and S2A) as provided by the manufacturer to detect immature myeloid cells. To avoid confusions all CD13-CD33+ were corrected to CD13+CD33+.
Reviewer 2 Report
In this study, the authors have evaluated the impact of IL3 in ex vivo expansion of long term HSCs.
Though, the results provided support the authors claim that IL3 in culture media has a negative effect on HSC self renewal property, it is concerning that the authors have used only one biological replicate to arrive at their conclusions. I would like to see their in vitro experiment repeated in CD34+ cells derived from atleast two additional independent human donors. Primary transplant experiments should be provided for atleast one additional donor. This is important to ascertain if the reported negative effects of IL3 has relevance beyond a single donor derived CD34 cells.
Author Response
We understand the reviewer’s concern, but rest assured that the conclusions are based on multiple cord blood donors and various different experiments. In each experiment, we have always used pooled samples from at least 3 to maximally 5 different donors for both in vitro and in vivo experiments. Previously we performed in vitro experiment in smaller scale in 96 well-half area plates and different seeding densities and the experiment originally shown was performed in full 96 well plates. However, we repeated the in vitro experiment as reviewer requested, using the very same exact culture conditions, starting with 100,000 cells of enriched CD34+ (Pooled from 3 different donors), and two biological replicates for each condition. Results from the new experiment were combined with the previous experiment, and Figure 1, Figure S1 BC are updated. (Corresponding to 6 different cod blood donors)
Reviewer 3 Report
In the article: “IL3 Has a Detrimental Effect on Hematopoietic Stem Cell Self-renewal in Transplantation Setting” the authors discussed about the negative effect of IL3 on HSC self-renewal.
Overall, this manuscript results very interesting, the authors clearly explain the rational of the study and discussed the topic point by point.
However, we would like to invite the authors to clarify some minor points:
1. Please check the check punctuation and spaces;
2. Among the introduction section, the authors describes in general the importance of HSC and insert some references. Please insert also a more recent reference in comparison to 1-3.
3. Among the introduction section the authors should deep the potential applications of stem cells and the possibility to induce a specific differentiation process. In this respect the following references should be useful: “Alessio N, Stellavato A, Aprile D, Cimini D, Vassallo V, Di Bernardo G, Galderisi U, Schiraldi C. Timely Supplementation of Hydrogels Containing Sulfated or Unsulfated Chondroitin and Hyaluronic Acid Affects Mesenchymal Stromal Cells Commitment Toward Chondrogenic Differentiation. Front Cell Dev Biol. 2021 Apr 12;9:641529. doi: 10.3389/fcell.2021.641529. PMID: 33912558; PMCID: PMC8072340.”
4. Why the used cells came from cord blood? Which passage of in vitro culture was used for the experiments?
5. Why IL-13 has been used at the concentration of 20ng/mL, another concentration has been tested?
Author Response
WE have made some of the recommended changes; The concentration of IL3 is the same as used in clinical protocols (see rely to reviewer 1). UCB is a common source of CD34+ HSCPs that can be readily obtained and is the standard source used in the field for this purpose.
Round 2
Reviewer 2 Report
Thanks for the clarification and the additional experiment.